# Chorus wave power at the strong diffusion limit overcomes electron losses due to strong diffusion

T. A. Daggitt ●[1,2] ✉, R. B. Horne ●[1], S. A. Glauert ●[1], G. Del Zanna ●[2] & J. M. Albert ●[3]

Earth's radiation belts consist of high-energy charged particles trapped by Earth's magnetic field. Strong pitch angle diffusion of electrons caused by wave-particle interaction in Earth's radiation belts has primarily been considered as a loss process, as trapped electrons are rapidly diffused into the loss cone and lost to the atmosphere. However, the wave power necessary to produce strong diffusion should also produce rapid energy diffusion, and has not been considered in this context. Here we provide evidence of strong diffusion using satellite data. We use two-dimensional Fokker-Planck simulations of electron diffusion in pitch angle and energy to show that scaling up chorus wave power to the strong diffusion limit produces rapid acceleration of electrons, sufficient to outweigh the losses due to strong diffusion. The rate of losses saturates at the strong diffusion limit, whilst the rate of acceleration does not. This leads to the surprising result of an increase, not a decrease in the trapped electron population during strong diffusion due to chorus waves as expected when treating strong diffusion as a loss process. Our results suggest there is a tipping point in chorus wave power between net loss and net acceleration that global radiation belt models need to capture to better forecast hazardous radiation levels that damage satellites.

Wave-particle interactions are an important driver of electron flux variations in the radiation belts, with local electron acceleration and loss driven by very-low frequency (VLF) waves including chorus, plasmaspheric hiss, lightning generated whistlers and electromagnetic ion-cyclotron (EMIC) waves[1-7]. Ultralow frequency (ULF) waves drive radial diffusion, accelerating electrons as they move inwards[8,9].

The diffusive effects of wave-particle interactions on radiation belt electrons are often modelled using quasilinear diffusion theory[10,11]. When considering only diffusion in pitch angle near the loss cone, Kennel[12] showed that high rates of pitch angle diffusion rapidly produce isotropic pitch angle distributions when electrons can be diffused across the loss cone on timescales close to one quarter of their bounce period. This results in the strong diffusion limit being defined as

$$\frac{1}{D_{SD}} = \frac{1}{4}\tau_B \tag{1}$$

where $D_{SD}$ is the pitch angle diffusion coefficient at the strong diffusion limit, and $\tau_B$ is the electron bounce period. For an electron with rest energy $E_0 = m_e c^2$, this can be approximated by[13]

$$D_{SD} = \frac{9.66}{L^4}\left(\frac{4L}{4L-3}\right)^{\frac{1}{2}}\frac{\left(E\left(E+2E_0\right)\right)^{\frac{1}{2}}}{E+E_0} \tag{2}$$

where E is the electron kinetic energy and L is the McIlwain L-shell[14].

[1]British Antarctic Survey, Cambridge, UK. [2]Department of Applied Maths and Theoretical Physics, University of Cambridge, Cambridge, UK. [3]Air Force Research Laboratory, Kirtland AFB, NM, USA. ✉e-mail: thoggi18@bas.ac.uk

Once the gradient of the pitch angle distribution is close to zero near the loss cone, increasing diffusion rates further will have minimal effect on the pitch angle distribution. This limit is known as the strong diffusion limit, and it represents the maximum rate of loss to the atmosphere under quasilinear theory.

VLF waves with amplitudes sufficient to cause pitch angle diffusion at or above the strong diffusion limit are often discussed as a mechanism leading to the depletion of MeV electron fluxes[15,16], and until 1998 scattering of electrons by VLF waves was only considered to result in electron losses[1]. However, as the pitch angle, energy and pitch angle/energy cross diffusion coefficients all increase proportionally with the square of the wave amplitude[17,18], high amplitude waves can cause both acceleration and pitch angle scattering[6]. Recent studies have demonstrated both rapid acceleration and loss caused by high amplitude chorus waves coupled with low plasma densities at different energies[19,20]. Determining the balance of loss and acceleration is important for forecasting the relativistic electron flux in order to minimise the risk to satellites from MeV electrons[21].

Here we find evidence of strong diffusion in electron pitch angle distributions and VLF wave observations. We demonstrate that strong diffusion due to chorus waves can result in rapid increases in electron flux at a range of energies. We also show that the balance of electron loss and acceleration due to chorus waves changes rapidly with increasing wave power leading to acceleration dominating over loss at MeV energies.

## Evidence for strong diffusion

Under strong diffusion the ratio of the precipitating directional, integral electron flux to the trapped flux should be close to unity as the velocity distribution becomes isotropic. Observations of the pitch angle distribution near the loss cone were derived from the Medium Energy Proton and Electron Detector (MEPED) instruments onboard Polar Orbiting Environmental Satellites (POES) M01 and N19 satellites, in polar orbit at an altitude of approximately 840 km. These instruments consist of two electron and two proton telescopes, pointed at 0° and 90° to local vertical. At geographic latitudes above 45° the 0° telescope points into the bounce loss cone and records count rates for precipitating electrons. At latitudes above 60° the 90° telescope primarily records trapped electrons with small equatorial pitch angles, which dominate over any counts that may be measured inside the bounce or drift loss cone by the 90° telescope[22,23]. Thus, at latitudes above 60° the ratio of fluxes derived from the count rates of each telescope gives a good indication of the gradient of the distribution across the loss cone. The effect of any overlap of the 90° telescope with the loss cone on the electron count rates is greatly reduced during strong diffusion[24]. Electron fluxes were derived from the MEPED data, including corrections for proton contamination of MEPED electron channels, using the method developed in Peck et al.[25].

Figure 1a shows the flux ratio from the POES satellites from 21:00 to 22:00 UTC during the 17th March 2013 St Patrick's day storm, which has previously been used as an example of high chorus wave power[20,26]. Here the ratio of the fluxes recorded in the >300 keV channel is close to unity for an extended period at high latitude, the ratio only drops significantly below one when the satellites move below L = 4, with a rapid transition near the typical plasmapause location. This transition suggests the presence of strong chorus waves outside the plasmapause that are causing rapid pitch angle diffusion. This transition was present in all storm events analysed during this study, however,

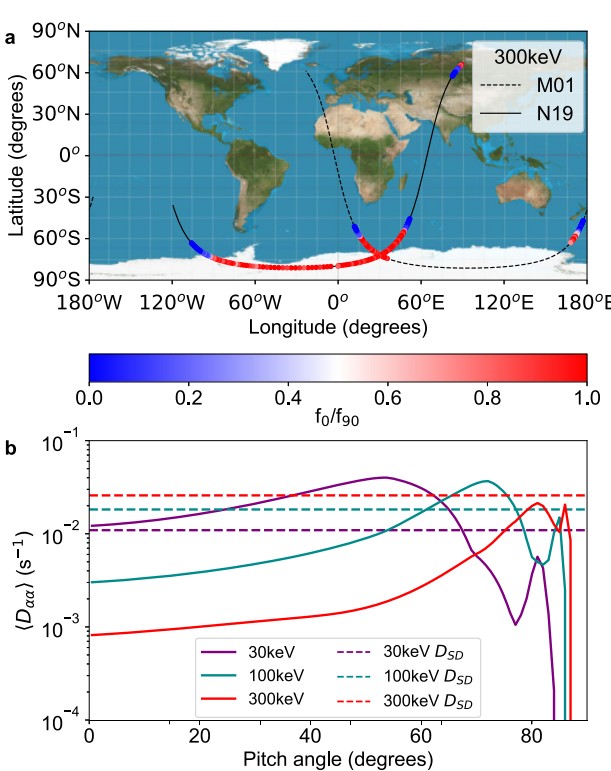

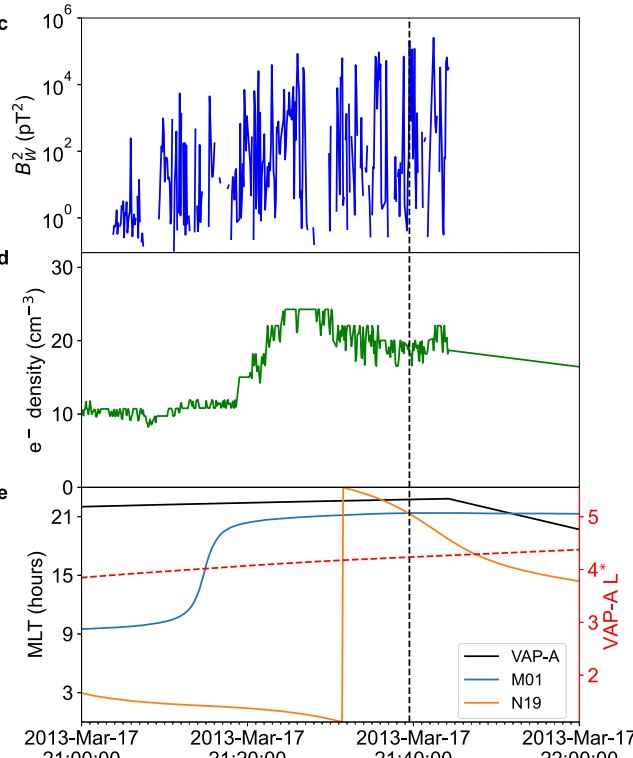

**Fig. 1 | VAP-A and POES M01 and N19 data from the 2013 St Patrick's day storm, 21:00 – 22:00 UTC. a** The ratio between the electron flux from the 0° and 90° telescopes for each POES satellites for the >300 keV channel along the satellite path (black lines) in geographic coordinates. **b** The pitch angle diffusion coefficients derived from the electron density and chorus wave power using quasilinear theory with the Pitch Angle and Energy Diffusion of Ions and Electrons (PADIE) code (see methods subsection 2. Diffusion Coefficients)[3] at 21:39:31UTC. The strong diffusion limits for each energy are indicated by the dashed horizontal lines. **c** The integrated lower-band chorus wave power at the VAP-A satellite. **d** the electron density at the VAP-A satellite, **e** L* at the **e** VAP-A satellite (red dashed line) as well as the magnetic local time at VAP-A and POES M01 and N19. The dashed vertical line in (**c**)–(**e**), indicates the time for which the diffusion coefficients in (**b**) were calculated. Source data are available from the UK Polar Data Centre[49].

observations during quiet times without low latitude chorus show no rapid change in the ratio of fluxes recorded by POES as crosses the plasmasphere (Supplementary Fig. 2). This shows that the pitch angle distribution is flat across the loss cone, implying that strong diffusion is occurring. The effects of field line curvature scattering can also result in POES observing flat pitch angle distributions[27], which can occur concurrently with scattering due to wave-particle interaction. As flat pitch angle distributions are observed here for 15 minutes, over a range of L-shells from 4 to 6, it is unlikely that field line curvature scattering alone can account for all the observations of flat pitch angle distributions in Fig. 1a (Supplementary Fig 1.). The classification tool of Capannolo et al.[27] detects both precipitation due to field line curvature scattering and wave-particle interactions in this event. The National Oceanic and Atmospheric Administration (NOAA) Space Environment Services Center lists a solar energetic particle (SEP) event finishing 14 h before the observations in Fig. 1. We assume the proton contamination correction leaves us able to correctly identify flat pitch angle distributions outside of SEP events, and the POES observations shown in Fig. 1a meet the requirement for low proton contamination specified in Rodger et al.[22].

The integrated lower band chorus wave power and electron density were calculated near the equator using survey mode data from the Electric and Magnetic Field Instrument Suite and Integrated Science (EMFISIS) instrument on the Van Allen Probes (VAP) A satellite, which orbits at an inclination of 10°. The chorus wave power was integrated from the lower hybrid resonance frequency to half the electron gyrofrequency. The density was calculated from the upper hybrid resonance frequency using the method of Kurth et al.[28]. The wave normal angle, $\psi$ was fitted to EMFISIS data using a Gaussian distribution in $tan(\psi)$[17,29], with a width of 12° (Supplementary Fig. 4). Whilst chorus waves occur as brief, intense wave bursts and are highly non-linear, causing particle trapping effects, we have used the average wave power from the EMFISIS instrument in survey mode, which is measured over a 0.5 s window every 6s[30], and applied quasilinear theory to calculate diffusion coefficients. This excludes the effects of particle trapping. Studies show that quasilinear and highly non-linear theory produce remarkable agreement for broad frequency band waves such as chorus after bounce-averaging the diffusion rates along a magnetic field line[31].

Figure 1c–e show VAP-A EMFISIS data during the period covered by the POES data. Bursts of high lower-band chorus wave power combined with low electron density were recorded by VAP-A whilst the satellite was at $L^* \approx 4$ (Fig. 1e), within the heart of the outer radiation belts. During this period the recorded chorus wave power was high enough to cause strong diffusion at 30 keV, as indicated by the diffusion coefficients in Fig. 1b exceeding the purple dashed line at small pitch angles. These coefficients were calculated from the observational data at 21:39:31 UTC, indicated by the vertical dashed line in Fig. 1c–e (see methods subsection 2. Diffusion Coefficients), with an equatorial squared chorus wave power of $2.03{\times}10^5$pT$^2$, near the peak of chorus wave power found in statistical studies of VAP observations during geomagnetic storms[20]. The pitch angle diffusion coefficients in 1B for the 100 keV and 300 keV energies fail to reach the strong diffusion limit at low pitch angles and thus will not result in a full loss cone. This is at odds with the flux ratio shown in Fig. 1a for the >300 keV channel, where the ratio is near to one for an extended period, showing that the pitch angle distribution is close to flat within the loss cone.

One possible reason for the discrepancy is that the pitch angle distribution measured by the POES satellites is the cumulative result of interactions with waves which map up along magnetic field lines to a large region of space near the magnetic equator, whereas VAP-A can only record wave power along its orbit near the equator. Although VAP-A can measure wave power on approximately the same field lines as the POES satellites, the data suggest that VAP-A did not observe the strongest waves at this time, or only observed part of the latitudinal

distribution of chorus wave power. Statistical studies of chorus waves have demonstrated the presence of high chorus wave power at equatorial latitudes and latitudes greater than 20° on the dayside during active periods[32,33], which VAP-A could not have observed during this event. The rate of loss of high energy electrons due to chorus waves is strongly affected by the presence of chorus wave power at high latitudes[34]. Additional interactions with chorus waves at higher latitudes along the field line where the background magnetic field is stronger may account for the flat pitch angle distributions seen in the >300 keV channel.

This event was chosen to illustrate concurrent strong chorus wave power and flat pitch angle distributions as evidence of chorus wave driven strong diffusion, a full statistical treatment of how often these conditions occur was not performed, and is beyond the scope of this study.

## Strong diffusion from chorus waves

Figure 2a shows the flux of electrons with an equatorial pitch angle of 85° from a simulation using the British Antarctic Survey 2D Radiation Belt Model[10] (BAS-RBM 2D). Initially diffusion is driven by statistical VLF wave models (see methods subsection 2. Diffusion Coefficients). At the 5-hour mark, the squared chorus wave power is scaled up by a factor of 2000 such that the pitch angle diffusion coefficient $D_{\alpha\alpha} = D_{SD}$ at 300 keV, without altering the values of the mixed pitch angle/energy and energy diffusion coefficients ($D_{\alpha E}$ and $D_{EE}$ respectively). This test is intended to recreate the assumptions under which the strong diffusion limit was originally specified, which did not consider energy diffusion significant[12]. This causes the pitch angle distribution gradient to approach zero across the loss cone at all energies below $E_{SD}$, the highest energy at which strong diffusion occurs, resulting in rapid losses. Energy diffusion is included, but the comparatively low rate of energy diffusion accelerates electrons up from the low energy boundary slower than they can be lost, resulting in loss of almost all flux above 150 keV. The flux above 1 MeV takes longer to decay, as $D_{\alpha\alpha}$ decreases with increasing energy. Once $D_{\alpha\alpha}$ is decreased to its original value after 19 h, the flux at lower energies begins to refill.

In quasilinear theory the proper effect of scaling up the chorus wave power should be to scale $D_{\alpha E}$ and $D_{EE}$ by the same factor as $D_{\alpha\alpha}$[17,18], the effect of this is shown in Fig. 2b. Here the additional energy diffusion results in a huge increase in the flux above 1 MeV. This is contrary to the original description of strong diffusion as a loss process. The loss of flux at lower energies is being partially offset by the increased acceleration of electrons up from the low energy boundary. At energies above 1 MeV the rate of acceleration begins to exceed the losses and results in increasing flux. Using a value of $E_{SD} = 30$ keV, scaling up the squared chorus wave power by a smaller factor of 150, results in lower values of $D_{EE}$, but the onset of strong diffusion still results in increasing flux for $E > 1$ MeV (Fig. 2c). The increase is negligible for energies closer to 10 MeV.

The rate of acceleration of electrons up to higher energies is determined by the phase-space density gradient in energy and the magnitude of the diffusion coefficients. Large gradients in energy are visible in Fig. 2b during strong diffusion, showing that the rate of energy diffusion has not reached a cap in the same manner as pitch angle diffusion. Scaling the diffusion coefficients up to reach $E_{SD} = 300$ keV causes order of magnitude increases in MeV fluxes within hours (Fig. 2b). In 2c strong diffusion is only reached for $E < 30$ keV. Here the acceleration at higher energies is much slower, taking close to 14 h to cause a significant increase in the flux at 3 MeV. The faster acceleration with increasing diffusion coefficients implies that high energy fluxes during active periods may be largely determined by higher levels of chorus wave power which may be averaged out in global radiation belt models. The length of these simulations is longer than the period of high chorus wave power shown in Fig. 1. The simulations are intended to show the steady state the system moves

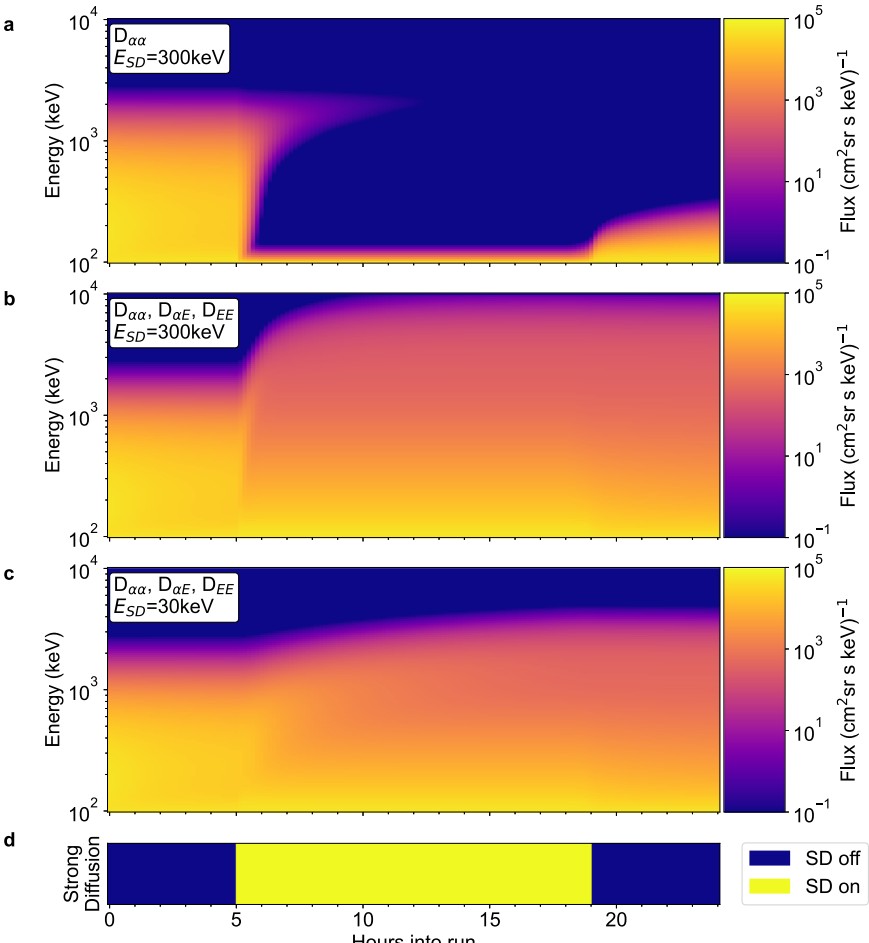

**Fig. 2 | Simulations of strong diffusion of electrons due to chorus waves.** 85° electron flux (colour coded) from BAS-RBM simulations plotted as a function of time and energy with an initial $\sin(\alpha)$ pitch angle distribution and an exponentially decaying energy spectrum. **a** The effect of scaling up the pitch angle diffusion coefficients, $D_{\alpha\alpha}$, reaching strong diffusion for electrons with $E = 300$ keV. **b** A simulation with $D_{\alpha\alpha}$, the energy diffusion coefficients, $D_{EE}$, and the mixed diffusion coefficients, $D_{\alpha E}$, scaled up to strong diffusion at $E = 300$ keV. **c** A simulation with $D_{\alpha\alpha}$, $D_{\alpha E}$ and $D_{EE}$ scaled up to strong diffusion at $E = 30$ keV. **d** Times coloured in blue or yellow show whether the strong diffusion scaling is turned on or off respectively. Source data are available from the UK Polar Data Centre[49].

towards under the influence of strong chorus wave power, and how fast that steady state is approached.

## Effects of chorus wave intensity on the steady state

The energies at which there is a net increase or decrease in flux depends on the initial and the boundary conditions of the run, and the chorus wave power. To investigate these effects, we ran simulations until steady state was reached for three energy spectra representing spectra observed before a typical magnetic storm, during the storm main phase and during the recovery phase[35]. The initial and final energy spectra for a chorus wave power scaling factor of $10^2$ are shown in Fig. 3a–c for a bump-on-tail, power law and exponential spectrum.

As the chorus scaling is increased, the ratio of final to initial phase-space density increases for all energies and for each spectrum (Fig. 3d–f). This implies that the rate of acceleration of trapped electrons increases faster than the rate of loss as chorus wave power increases. As the values of $D_{\alpha\alpha}$, $D_{\alpha E}$ and $D_{EE}$ are scaled by the same value, this difference must be due to the gradients of flux in pitch angle and energy. Fig. 3a–c show that although the gradient becomes smaller, large energy gradients remain after reaching steady state under the influence of strong diffusion.

The effect of the initial energy spectrum on whether chorus waves cause an increase or decrease in flux can be seen by comparing Fig. 3d–f. Fig. 3d uses a bump-on-tail distribution, comprised of a power law with a gaussian added at higher energies, to represent the effects of chorus on a pre-storm energy spectrum. At 300 keV the ratio of the final to initial phase-space density increases slowly with chorus scaling, starting close to unity without scaling (Fig. 3d). By contrast, the ratio at 3 MeV starts much smaller and increases much more rapidly with chorus scaling, moving from a net loss to a net increase at chorus scaling factors >$10^2$. Fig. 3e uses a power law distribution, which can occur during the main phase of a storm due to injections of low energy electrons. The steep energy gradient at all energies results in acceleration outweighing losses to the atmosphere at most energies and levels of chorus scaling. The effects of chorus waves on energy spectra during the recovery phase is shown in Fig. 3f, using an exponential distribution. Losses dominate even for extreme values of chorus wave power at the lower energies as the phase-space density energy gradient is small.

Figure 3e, f all show that the ratio of the final to initial phase-space density at 3 MeV increases faster with chorus scaling than the ratio at lower energies. This shows that higher energies are more sensitive to increasing chorus wave power. A factor of 30 in the squared chorus wave power is enough to move from decreasing to increasing phase-space density at 3 MeV for all three spectra shown in Fig. 3.

## Discussion

The chorus pitch angle diffusion coefficients shown in section 1 were calculated from electron density and chorus wave power observed by

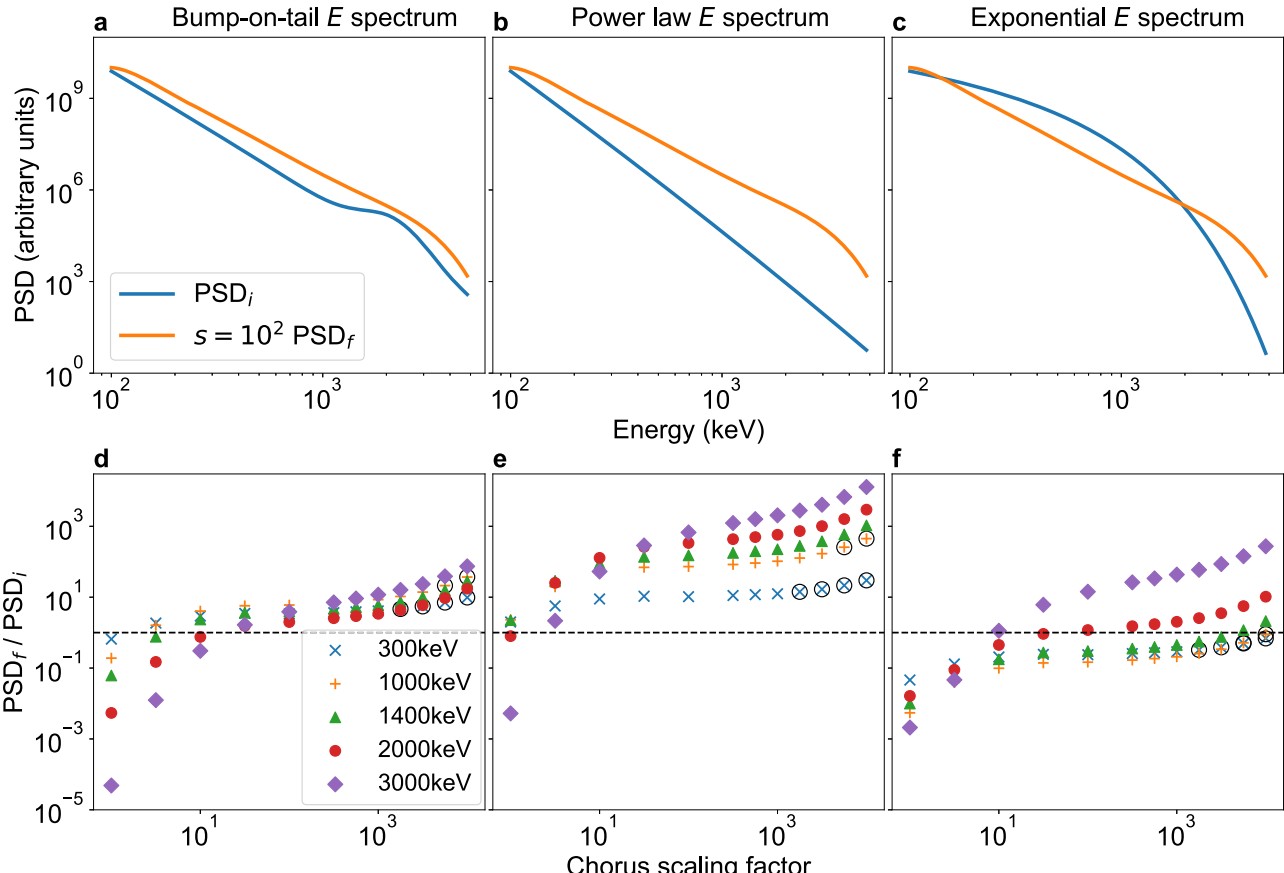

**Fig. 3 | The effect of changing chorus wave power on the steady-state electron phase-space-density. a–c** The top row shows the initial electron energy spectrum (PSD$_i$) and the energy spectrum at steady state (PSD$_f$) at 85° pitch angle for simulations using a chorus scaling factor of $s = 10^2$, for different initial energy spectra. **d–f** The bottom row shows the ratio of the final to initial phase-space density at for different chorus diffusion scaling factors, for simulations run until they reached a steady state for each initial energy spectrum. The circled points show ratios for which the pitch angle diffusion was greater than the strong diffusion limit. Source data are available from the UK Polar Data Centre[49].

the EMFISIS instrument in survey mode, and are 66 times larger than the diffusion coefficients near the loss cone calculated at the corresponding location from the wave model of Meredith et al.[36], and 1500 times larger for near-equatorially mirroring electrons. This difference is due to averaging over many observations when creating statistical models of wave activity, which results in diffusion coefficients much smaller than the extremes seen in single observations. As the high energy electron fluxes may be sensitive to short-lived peaks in chorus wave power, statistical models of wave activity may result in poor recreation of the behaviour of high energy fluxes in radiation belt models during storm times. The diffusion coefficients in Fig. 1 were recalculated a number of times using different wave normal angle distributions, and were found to be nearly independent of wave normal angle distribution for pitch angles below 80°.

Figure 3 suggests that there is a tipping point in chorus wave power between a decrease in high energy fluxes and an increase that global radiation belt models need to capture to better forecast hazardous radiation levels that damage satellites. A small underestimation in chorus wave power may lead to a simulation showing a decrease in high energy fluxes instead of an increase under the correct circumstances. This suggests that global radiation belt simulations using averaged chorus wave power may misrepresent the rate of loss and acceleration due to chorus waves and should be revised.

All simulations have been performed using a constant low energy boundary to represent a fixed population of low energy electrons. Previous work has demonstrated that the source and seed populations of electrons are highly variable during storms and substorms[37,38]. This

has been shown to affect the fluxes of high energy electrons in simulations and observations[19,37]. Altering the phase-space density on the low energy boundary changes the balance of loss and acceleration, as it will alter the phase-space density energy gradient. The effects of different energy gradients at low energies are demonstrated in Fig. 3. An injection of low energy electrons will result in more chorus wave growth and an increase in the phase-space density energy gradient. This will lead to more rapid acceleration, enhancing flux at higher energies. Conversely, losing low energy electrons will decrease acceleration, resulting in chorus causing dropouts at higher energies. The sensitivity of high energy fluxes to changes in $D_{EE}$ suggests that the effects of variability in the low energy boundary will be most extreme during periods with high chorus wave power. The low energy boundary was placed at 100 keV to ensure that the effects of the convection electric field can be ignored[39]. Altering the position of the low energy boundary has a similar effect on the steady state energy spectrum to raising or lowering the phase space density at the existing low energy boundary (Supplementary Fig. 3).

We found evidence of strong diffusion occurring within the magnetosphere as indicated by flat pitch angle distributions near the loss cone observed in low Earth orbit, and chorus wave power sufficient to reach the strong diffusion limit at 30 keV near the equator. These data were observed simultaneously using data from the VAP-A and POES satellites. Evidence of flat pitch angle distributions was observed up to 300 keV.

Simulations with the BAS-RBM 2D showed the surprising result that acceleration of electrons from a fixed low energy seed population can overcome losses due to strong diffusion, demonstrating that the

rate of acceleration is not capped in the same manner as the rate of loss. This results in rapid increases in high energy fluxes when chorus wave power is high.

The model predicts that the flux at MeV energies is much more sensitive to chorus wave power than the flux at hundreds of keV. A factor of 30 increase in the values of an existing chorus diffusion model is sufficient to move from losing flux to gaining flux at MeV energies.

We find that the balance of loss and acceleration due to chorus wave activity is also dependent on the energy spectrum that it is acting upon. Acceleration is liable to dominate during the main phase of a storm, but is outweighed by losses pre-storm and during the recovery, except for very high chorus activity.

Much current work modelling the radiation belts using the quasilinear theory of diffusion, including the BAS-RBM, relies on diffusion coefficients that are bounce- and drift-averaged, calculated from wave models that are themselves averaged[10,19,34,40]. This may fail to capture the behaviour of electrons at MeV energies during highly active periods, as the evolution of high energy populations will be strongly affected by where the average chorus wave power lies with respect to the tipping point. Due to this, global radiation belt models using these averaged wave models may underestimate the rate of loss and acceleration due to chorus waves, resulting in underestimations of the changes in flux during storms. The way in which the wave power and or the diffusion rates are averaged can make a significant difference[40,41].

The results here are applicable to other planets with magnetospheres such as Jupiter, Saturn, Neptune and Uranus, where strong diffusion is likely to occur, and to magnetically confined plasmas in the laboratory.

## Methods

### The 2D BAS radiation belt model

The stochastic behaviour of phase-averaged electron phase-space density, $f$, distributions can be described by a Fokker-Planck diffusion equation[8]. The 2D British Antarctic Survey Radiation Belt Model[10] (BAS-RBM 2D) considers particles at a single $L^*$ without radial diffusion, and solves the equation transformed to pitch angle, $\alpha$ and energy, $E$ space, including cross terms given by

$$\frac{\partial f}{\partial t} = \frac{1}{g(\alpha)} \frac{\partial}{\partial \alpha}\bigg|_E g(\alpha)\left(D_{\alpha\alpha}\frac{\partial f}{\partial \alpha}\bigg|_E + D_{\alpha E}\frac{\partial f}{\partial E}\bigg|_\alpha\right) \\ + \frac{1}{A(E)}\frac{\partial}{\partial E}\bigg|_\alpha A(E)\left(D_{EE}\frac{\partial f}{\partial E}\bigg|_\alpha + D_{E\alpha}\frac{\partial f}{\partial \alpha}\bigg|_E\right) - \frac{f}{\tau} \tag{3}$$

where

$$g(\alpha) = \sin(\alpha)\cos(\alpha)T(\alpha) \tag{4}$$

$$A(E) = (E + E_0)(E(E + 2E_0))^{\frac{1}{2}} \tag{5}$$

and $E_0$ is the electron rest energy. $T(\alpha)$ is related to the electron bounce period, and arises from the bounce-averaging of the diffusion coefficients. Losses to the atmosphere are accounted for by the loss term $f/\tau$, with the loss timescale $\tau$ set to a quarter bounce period within the loss cone and infinite elsewhere. $D_{\alpha\alpha}$, $D_{E\alpha} = D_{\alpha E}$ and $D_{EE}$ are the bounce- and drift-averaged pitch angle, cross and energy diffusion coefficients. Future references to these terms in this study are to be assumed bounce- and drift-averaged unless explicitly specified. The drift-averaged coefficients were calculated by averaging the bounce-averaged diffusion coefficients across all magnetic local time (MLT) sectors. This will result in lower peak values than the bounce-averaged coefficients, but is a common simplifying assumption used in radiation belt models[10,42]. Equation 3 is solved, including the effects of pitch angle-energy cross diffusion using the Hunsdorfer-Verwer method, an alternating implicit difference solver[43], with nine-point discretisation.

The use of this method within the BAS-RBM 2D has previously been described by Woodfield et al.[44].

The BAS-RBM and BAS-RBM 2D have previously been used in multiple studies and have successfully reproduced a range of radiation belt phenomena[10,44,45].

### Diffusion coefficients

The BAS-RBM 2D represents the effect of wave-particle interactions within the magnetosphere using diffusion coefficients computed from wave models. Separate models for chorus, plasmaspheric hiss and EMIC waves have been published by Meredith et al.[36], Glauert et al.[10] and Ross et al.[40] and bounce- and drift-averaged diffusion coefficients derived from these using quasilinear theory[10] have been included in the BAS-RBM 2D. To capture the effect of geomagnetic activity on the bounce- and drift-averaged diffusion coefficients, the diffusion coefficients for each of these waves have been binned by geomagnetic activity, using the $Kp$ index. As the conditions required for strong diffusion do not occur in the chorus diffusion model, in order to study the effects of strong diffusion, we have applied a scale factor to the chorus diffusion coefficients, increasing $D_{\alpha\alpha}$, $D_{\alpha E}$ and $D_{EE}$ until $D_{\alpha\alpha} = D_{SD}$ at a specific energy. Although the chorus diffusion model does not reach the strong diffusion limit, strong diffusion may occur in nature on short timescales, observations of which may be included in the averaged wave power use to form the diffusion coefficients used in the BAS-RBM 2D.

The observation specific pitch angle diffusion coefficients for different energies shown in Fig. 1b were derived from measurements of electron number density and the frequency distribution of lower band chorus wave power. Fully relativistic quasi-linear pitch angle and energy diffusion coefficients from resonant wave-particle interactions were calculated using the full electromagnetic dispersion relation within a magnetised plasma, using the PADIE code[17]. Calculations were performed assuming a Gaussian wave normal distribution fitted to the distribution of $\tan(\psi)$, where $\psi$ is the wave normal angle from the VAP-A EMFISIS instrument L4 data, with a width of 12º (Supplementary Fig. 3), and waves extending to a magnetic latitude of 20º, based on the observed magnetic latitude distributions in Meredith et al.[36]. The calculation included the resonance numbers from −10 to 10, including the Landau resonance, in an electron-proton plasma. The PADIE code has been compared to a particle-in-cell approach[46], and used to calculate diffusion coefficients for a variety of waves including chorus[47], hiss[10] and EMIC waves[40].

### Boundary conditions

The BAS-RBM 2D solves Eq. 3 on a fixed grid of energies and pitch angles at a single $L^*$, with boundary conditions supplied at the minimum and maximum energy and pitch angle values, and an initial condition at $t = 0$. The simulations in this study have used $\partial f/\partial\alpha = 0$ boundary conditions at 0º and 90º. The low energy boundary is placed at $E_{min} = 100$ keV, with $f(\alpha, E_{min}) = \sin(\alpha)f(90, E_{min})$. The pitch angle distribution at the low energy boundary is based on $\sin^n(\alpha)$ distributions fitted to data[48] at $Kp = 4$ and $E = 100$ keV, where n = 1. The high energy boundary is placed at $E_{max} = 10$ MeV, with $f(E_{max}) = 0$. As the BAS-RBM 2D uses the logarithm of the phase space density, $f(E_{max}) = 0$ is treated as a very small non-zero value. The initial condition for Fig. 2 is chosen to match the low and high energy boundaries, and assumes an exponential decay in phase-space density with energy. The shape of the initial spectra for each simulation in Fig. 3 are shown in the figure. Low values of phase space density in the initial condition below the small non-zero value at the high energy boundary are set equal to the phase space density on the high energy boundary, with minimal effect on the model output.

These conditions were chosen to be easily reproducible, representative of typical distributions in the radiation belts[35,48] at $L^* = 4.5$, and relatively stable under the influence of the unmodified diffusion coefficients due to chorus, hiss and EMIC waves detailed earlier. The shape of the pitch angle distribution has little effect on the long-term

evolution of electron phase-space density as the distribution flattens over timescales close to the bounce period of an electron near the loss cone under strong diffusion.

## Data availability

The POES MEPED data from NOAA National Geophysical Data Centre for the M01 and N19 satellites are available at https://www.ncei.noaa.gov/data/poes-metop-space-environment-monitor/access/l1a/v01r00/2013/metop01/ and https://www.ncei.noaa.gov/data/poes-metop-space-environment-monitor/access/l1a/v01r00/2013/noaa19/. The EMFISIS data from the NASA Van Allen Probes are available at https://emfisis.physics.uiowa.edu/Flight/RBSP-A/L4/2013/03/17/. Source data are provided with this paper for the diffusion coefficients and BAS-RBM 2D simulation data generated in this study, as well as the POES and VAP-A data shown in the study. The files are freely available from the NERC EDS UK Polar Data Centre database[49] at https://doi.org/10.5285/c7db6003-1f72-4e69-b0b4-ec4b0aa4763c. Source data are provided with this paper.

## Code availability

The BAS-RBM 2D and PADIE codes are available from the authors (S.A.G. and R.B.H. via sagl@bas.ac.uk and rh@bas.ac.uk) on request for the purpose of replicating our findings. We have chosen not to publicly share the code in an online repository in order to protect proprietary algorithms. We are committed to fostering scientific collaboration and are happy to provide the code to interested researchers for the purpose building upon our findings in a joint research collaboration.

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

## Acknowledgements
This material is based upon work supported by the Air Force Office of Scientific Research under award number FA9550-19-1-7039. R. B. Horne and S. A. Glauert were also supported by the Natural Environment Research Council (NERC) grant NE/V00249X/1 (Sat-Risk) and National and Public Good activity grant NE/R016445/1. G. Del Zanna acknowledges support from STFC (UK) via the consolidated grants to the astrophysics group at DAMTP, University of Cambridge (ST/P000665/1 and ST/T000481/1). J. Albert acknowledges support from NASA Grant No. 80NSSC20K1270.

## Author contributions
T.A.D. wrote the manuscript, analysed the POES and VAP data, and performed the simulations using the BAS-RBM 2D. S.A.G. wrote the BAS-RBM 2D code. J.A. advised on the experimental methodology. All authors participated in the interpretation of the data and read and commented on the manuscript. R.B.H., S.A.G. and G.D.Z. supervised the work.

## Competing interests
The authors declare no competing interests.
