## [Peer Review File · Nature Communications]

REVIEWER COMMENTS

Reviewer #1 (Remarks to the Author):

Review report of the manuscript "Electron Acceleration During Strong Pitch Angle Diffusion" by Daggitt et al.

This is an interesting and important study investigating the possibilities of accelerating electrons in the strong pitch-angle diffusion regime. Strong pitch-angle diffusion has been regarded as a fast loss mechanism of energetic electrons in the magnetosphere. This study points out that even in the strong diffusion regime, the net effect of chorus waves on MeV electrons can be acceleration. This point has been implicitly included in 3D simulations including radial diffusion, pitch-angle diffusion, energy diffusion and mixed energy and pitch-angle diffusion. However, previous studies did not include strong diffusion at ~ 300 keV. For example, for the event considered in this draft, Shprits et al (2015) and Wang et al (2020) reproduced the MeV electron acceleration using simulations including radial diffusion, pitch-angle diffusion, energy diffusion and mixed energy and pitch-angle diffusion, but these studies did not include strong diffusion at 300 keV. By looking at the precipitating electrons measured by POES satellite, this study shows that 300 keV electrons reach the strong diffusion regime. Furthermore, they showed that even in this situation, the net effect of chorus waves on MeV electrons can be acceleration. To make this paper clearer to readers, I suggest the authors to consider addressing the following comments and questions.

Major comments (addressing these comments will not change the main conclusion but will help to make the paper clearer):

1. Lines 77-78: at which L-shells are the POES satellite? Did the author consider eliminating the effect of field line curvature scattering? I ask this because sometimes these "strong diffusion" cases are caused by the field line curvature scattering rather than wave-particle interactions. See figure 1 of Capannolo et al (2022).

Capannolo L, Li W and Huang S (2022) Identification and Classification of Relativistic Electron Precipitation at Earth Using Supervised Deep Learning. *Front. Astron. Space Sci.* 9:858990. doi: 10.3389/fspas.2022.858990

2. The lower energy boundary condition of the simulations is set at 100 keV with constant phase space density. Does this mean that you have a constant source at 100 keV? It is a bit difficult to have

constant phase space density at 100 keV. It would be interesting to test the simulations with a lower energy boundary at 10 keV.

Minor comments:

1. Line 42: is E the electron 'kinetic' energy?
2. Please consider adding a reference here to show 14 hours gap is enough to eliminate the influence of SEP
3. Line 89: mention the method of calculating diffusion coefficients here
4. Lines 93-97: are the authors using EMFISIS L4 data? I remember the time resolution of EMFISIS survey mode wave data is 6 seconds.
5. Line 97: What amplitude is the averaged wave power? Can this amplitude sustain several hours as the authors used in their simulations?
6. Line 118: I agree. Van Allen Probe A was on the nightside and it orbits magnetic latitude lower than 20 degrees. This makes it hard to see the strong chorus waves on the dayside high latitude. Authors can cite here statistical studies of chorus waves (e.g., Meredith et al., 2012; Wang et al., 2019) to make it clearer. Figure 12i of Wang et al (2019) showed that the intensity of lower band chorus waves on the dayside increase with latitude. Then Wang and Shprits (2019) showed the importance of chorus waves at high latitude.

Meredith, N. P., Horne, R. B., Sicard-Piet, A., Boscher, D., Yearby, K. H., Li, W., and Thorne, R. M. (2012), Global model of lower band and upper band chorus from multiple satellite observations, *J. Geophys. Res.*, 117, A10225, doi:10.1029/2012JA017978.

Wang, D., Shprits, Y. Y., Zhelavskaya, I. S., Agapitov, O. V., Drozdov, A. Y., & Aseev, N. A. (2019). Analytical chorus wave model derived from Van Allen Probe observations. *Journal of Geophysical Research: Space Physics*, 124, 1063–1084. <https://doi.org/10.1029/2018JA026183>

Wang, D., and Shprits, Y. Y. (2019). On how high-latitude chorus waves tip the balance between acceleration and loss of relativistic electrons. *Geophysical Research Letters*, 46, 7945–7954. <https://doi.org/10.1029/2019GL082681>

7. Line 127: this is an interesting test, but it is also unrealistic. Daa, DaE and DEE have the same scaling with Bw^2 . I understand authors would like to test the situation when Daa is strong while DaE and DEE are relatively weaker. Please elaborate more.

8. Line 175: DEE (space here) are
9. Line 191: figure 3D?
10. Line 204: is 30 the scaling factor of Bw or Bw²?
11. Line 386: no loss term in the equation (1)

Reviewer #2 (Remarks to the Author):

This paper describes simulations of electron diffusion using the 2D BAS Radiation Belt Model, in particular associated with strong diffusion induced by chorus waves. I believe the results are correct and to be of interest to radiation belt modellers. This submission is more of a technical note than a scientific paper. As such, it would not be appropriate for GRL or JGR(Space Physics). However, this submission may be appropriate for Nature Communications.

Reviewer #3 (Remarks to the Author):

Review report on manuscript

"Electron Acceleration During Strong Pitch Angle Diffusion"

by T. A. Daggitt et al. submitted to Nature Communications

Reviewed by O. Santolik, 25 July 2023

The manuscript is based on results of quasi-linear calculations of diffusion coefficients of energetic electrons in the outer Van Allen radiation belt. The authors scale all the diffusion

coefficients up to the strong diffusion limit, and show that energy diffusion to relativistic energies can outweigh losses caused by pitch angle diffusion.

Although the main result is obtained by theoretical calculations using simplified models, experimental data from the POES and Van Allen Probes spacecraft are also used in order to support the theory.

Generally, the text is well written and easily readable.

Problems I see in it are mainly linked to experimental data, as detailed below in my specific comments. Nevertheless, the logic of the main message is so straightforward that it makes me wonder why it hasn't been brought up earlier.

This is probably a sign of a good paper and I therefore recommend it for publication in Nature Communications after a revision is done by the authors.

Specific comments:

line 59... The condition of equal precipitated flux to the trapped flux seems unclear as it is written here, although the context shows that it probably just means that the velocity distribution has to be isotropic.

I recommend to reword the sentence and/or properly define the meaning of "flux" in this context.

line 78... I recommend to explain why the flux ratio

suddenly switches from ~ 0 to ~ 1 at a latitude of 50-60 deg.

Coordinates should also be defined, I assume these are geographic (not geomagnetic) coordinates. I also recommend to show that this is different from other cases, where the strong diffusion is absent. One summary sentence in the main text and a well documented set of cases in the Supplementary material would be good to prove the point and this would also certainly fit in the format of Nature Communications.

line 93... The frequency band over which the measured power spectral density was integrated should be mentioned.

line 93 bis...

The B^2 weighted average of wave vector directions can also be easily obtained from the EMFISIS Waves survey mode data, to verify the assumption of a Gaussian wave normal distribution mentioned on line 417. Again, this can be summarized by one sentence in the main text, with a reference to the technique, and some Supplementary material, all of which still fits in the format of Nature Communications.

line 93 tris....

Different techniques for estimation of the plasma density exist in the literature and it should be said here which of them is used. A reference would cure it.

line 120 ... Figure 1B shows that the equatorial squared amplitude would need to be at least 30-50 time larger to in order to reach the theoretical strong diffusion limit in a broader interval of pitch angles for 300 keV electrons.

I recommend the authors to specifically state what squared amplitudes are they actually using for their calculation.

From Fig 1C I can guess that it is probably around 10^5 nT^2 .

This leads to chorus amplitudes above 1nT for reaching strong diffusion at 300 keV. Such amplitudes have been observed by Van Allen Probes data but this should also be supported by reference(s).

An alternative explanation can be that the interaction takes place at higher latitudes beyond the Van Allen Probes orbit, where the background magnetic field is stronger, modifying the resonance condition.

line 126 ... I suggest to mention the scaling factor here

line 130 ... One can guess from the context, but a few words to define E_{SD} at the first occurrence of this symbol would make the text better readable.

line 158 ... I recommend to explicitly state that scaling factor for 300 keV is higher than for 30 keV (which should also be given here)

line 191... 1D -> 3D ?

line 210... I see a discrepancy here: A factor of 1500 is observed between the model diffusion coefficients and the one calculated from measured waves amplitudes. Yet, according to Fig 1B, at 300 keV we still are by a factor of 30-50 below the strong diffusion limit. It means that a factor of at least 45000 against the model is needed to reach the limit. The problem is that in Fig 3D-F the circled symbols for 300 keV start at a factor of 2000. Therefore I have a problem of more than one order of magnitude here. I guess that I'm just missing something in this estimate but I recommend the authors to add an explanation to the text.

line 417... I recommend to repeat what is the width of the Gaussian distribution in the model

line 430... The initial exponential decay of the phase space density might be better described, it seems inconsistent with PSD=0 at 10 MeV and also with the initial PSDs in fig3.

REPLIES TO REVIEWER COMMENTS

Replies are made immediately after each comment, formatted in *green italics*. Thank you to all the reviewers for their inputs.

REVIEWER COMMENTS

Reviewer #1 (Remarks to the Author):

Review report of the manuscript "Electron Acceleration During Strong Pitch Angle Diffusion" by Daggitt et al.

This is an interesting and important study investigating the possibilities of accelerating electrons in the strong pitch-angle diffusion regime. Strong pitch-angle diffusion has been regarded as a fast loss mechanism of energetic electrons in the magnetosphere. This study points out that even in the strong diffusion regime, the net effect of chorus waves on MeV electrons can be acceleration. This point has been implicitly included in 3D simulations including radial diffusion, pitch-angle diffusion, energy diffusion and mixed energy and pitch-angle diffusion. However, previous studies did not include strong diffusion at ~300 keV. For example, for the event considered in this draft, Shprits et al (2015) and Wang et al (2020) reproduced the MeV electron acceleration using simulations including radial diffusion, pitch-angle diffusion, energy diffusion and mixed energy and pitch-angle diffusion, but these studies did not include strong diffusion at 300 keV. By looking at the precipitating electrons measured by POES satellite, this study shows that 300 keV electrons reach the strong diffusion regime. Furthermore, they showed that even in this situation, the net effect of chorus waves on MeV electrons can be acceleration. To make this paper clearer to readers, I suggest the authors to consider addressing the following comments and questions.

Major comments (addressing these comments will not change the main conclusion but will help to make the paper clearer):

1. Lines 77-78: at which L-shells are the POES satellite? Did the author consider eliminating the effect of field line curvature scattering? I ask this because sometimes these "strong diffusion" cases are caused by the field line curvature scattering rather than wave-particle interactions. See figure 1 of Capannolo et al (2022).

Capannolo L, Li W and Huang S (2022) Identification and Classification of Relativistic Electron Precipitation at Earth Using Supervised Deep Learning. *Front. Astron. Space Sci.* 9:858990. doi: 10.3389/fspas.2022.858990

The POES satellites cross a range of L-shells from 4 to 6. This has been specified in line 86, and is visible in the supplementary material fig. 1 for the revised manuscript.

The effect of field line curvature scattering was not initially considered. The manuscript now acknowledges the effects of field line curvature scattering, and briefly presents evidence of both effects occurring, with an additional plot in the style of figure 1 of Cappannolo et al (2022) in the supplementary material.

2. The lower energy boundary condition of the simulations is set at 100 keV with constant phase space density. Does this mean that you have a constant source at 100 keV? It is a bit difficult to have constant phase space density at 100 keV. It would be interesting to test the simulations with a lower

energy boundary at 10 keV.

The constant low energy boundary at 100keV does act as a constant source at 100keV. We have included further discussion of the energy at which the low energy boundary condition is set has been included at the end of the discussion section (line 144), as well as including plots showing the effect of a 10keV low energy boundary in the supplementary material fig. 3. Moving the low energy boundary has a similar effect on the steady state distribution to simply changing the PSD at the boundary at 100keV.

Minor comments:

1. Line 42: is E the electron 'kinetic' energy?

Yes, we have now defined E as electron kinetic energy in the revised manuscript.

2. Please consider adding a reference here to show 14 hours gap is enough to eliminate the influence of SEP

We have now added a clause specifying that the POES observations shown in figure 1 meet the low proton contamination tests given in Rodger et al. (2010), with a citation (line 93). Upon review we felt that this was a more rigorous way to show that proton contamination should not be an issue.

3. Line 89: mention the method of calculating diffusion coefficients here

We have now stated in the caption that we have used quasilinear theory and the PADIE code for the calculation of diffusion coefficients (line 101), with a reference to the more detailed methods section.

4. Lines 93-97: are the authors using EMFISIS L4 data? I remember the time resolution of EMFISIS survey mode wave data is 6 seconds.

We are using EMFISIS L4 data. Survey mode data is recorded every 6s, based on a 0.5s sample window, Kletzing et al. (2013). This has been clarified and has been cited in the manuscript (line 115).

5. Line 97: What amplitude is the averaged wave power? Can this amplitude sustain several hours as the authors used in their simulations?

The averaged wave power used for the diffusion coefficient calculation was $0.2nT^2$. We have now stated this in the revised manuscript (line 130).

We do not intend to present evidence that this amplitude can be sustained for the length of the simulation, and our opinion is that this is unlikely. We have now clarified in the revised manuscript that the simulations are not an attempt to reproduce any particular event and are only intended to show the effects on steady state produced by high chorus wave power (i.e. acceleration dominating over loss) (line 198).

6. Line 118: I agree. Van Allen Probe A was on the nightside and it orbits magnetic latitude lower than 20 degrees. This makes it hard to see the strong chorus waves on the dayside high latitude. Authors can cite here statistical studies of chorus waves (e.g., Meredith et al., 2012; Wang et al., 2019) to make it clearer. Figure 12i of Wang et al (2019) showed that the intensity of lower band chorus waves on the dayside increase with latitude. Then Wang and Shprits (2019) showed the importance of chorus waves at high latitude.

Meredith, N. P., Horne, R. B., Sicard-Piet, A., Boscher, D., Yearby, K. H., Li, W., and Thorne, R. M. (2012), Global model of lower band and upper band chorus from multiple satellite observations, *J. Geophys. Res.*, 117, A10225, doi:10.1029/2012JA017978.

Wang, D., Shprits, Y. Y., Zhelavskaya, I. S., Agapitov, O. V., Drozdov, A. Y., & Aseev, N. A. (2019). Analytical chorus wave model derived from Van Allen Probe observations. *Journal of Geophysical Research: Space Physics*, 124, 1063–1084. <https://doi.org/10.1029/2018JA026183>

Wang, D., and Shprits, Y. Y. (2019). On how high-latitude chorus waves tip the balance between acceleration and loss of relativistic electrons. *Geophysical Research Letters*, 46, 7945–7954. <https://doi.org/10.1029/2019GL082681>

We have added additional discussion of the dayside chorus wave power and high latitude chorus wave power which may contribute to the discrepancy between the POES data and the waves seen by the VAP (line 144).

7. Line 127: this is an interesting test, but it is also unrealistic. Daa, DaE and DEE have the same scaling with Bw2. I understand authors would like to test the situation when Daa is strong while DaE and DEE are relatively weaker. Please elaborate more.

Our intention was to simulate a situation similar to the original Kennel 1969 formulation of the strong diffusion limit, where energy diffusion was considered to be insignificant. We have added a sentence to the revised manuscript explaining this (line 165), and another clarifying that the tests scaling Daa, DaE and DEE by the same factor are more realistic given modern theory.

8. Line 175: DEE (space here) are

There was a space, but it was formatted as subscript and italic, and thus smaller. This has been fixed.

9. Line 191: figure 3D?

The wrong figure panel was referenced, we have now corrected it to say figure 3D.

10. Line 204: is 30 the scaling factor of Bw or Bw2?

We have clarified in the revised manuscript that we are referring to a factor of 30 in the squared wave power (line 249).

11. Line 386: no loss term in the equation (1)

There was a missing loss term, we have now added it into the revised manuscript.

Reviewer #2 (Remarks to the Author):

This paper describes simulations of electron diffusion using the 2D BAS Radiation Belt Model, in particular associated with strong diffusion induced by chorus waves. I believe the results are correct and to be of interest to radiation belt modellers. This submission is more of a technical note than a scientific paper. As such, it would not be appropriate for GRL or JGR(Space Physics). However, this submission may be appropriate for Nature Communications.

Reviewer #3 (Remarks to the Author):

Review report on manuscript
"Electron Acceleration During Strong Pitch Angle Diffusion"
by T. A. Daggitt et al. submitted to Nature Communications

Reviewed by O. Santolik, 25 July 2023

The manuscript is based on results of quasi-linear calculations of diffusion coefficients of energetic electrons in the outer Van Allen radiation belt. The authors scale all the diffusion coefficients up to the strong diffusion limit, and show that energy diffusion to relativistic energies can outweigh losses caused by pitch angle diffusion. Although the main result is obtained by theoretical calculations using simplified models, experimental data from the POES and Van Allen Probes spacecraft are also used in order to support the theory.

Generally, the text is well written and easily readable. Problems I see in it are mainly linked to experimental data, as detailed below in my specific comments. Nevertheless, the logic of the main message is so straightforward that it makes me wonder why it hasn't been brought up earlier. This is probably a sign of a good paper and I therefore recommend it for publication in Nature Communications after a revision is done by the authors.

Specific comments:

line 59... The condition of equal precipitated flux to the trapped flux seems unclear as it is written here, although the context shows that it probably just means that the velocity distribution has to be isotropic.

I recommend to reword the sentence and/or properly define the meaning of "flux" in this context.

We have now specified that we are referring to direction, integral electron flux, and that this condition arises because the velocity distribution becomes isotropic under strong diffusion (line 59).

line 78... I recommend to explain why the flux ratio suddenly switches from ~ 0 to ~ 1 at a latitude of 50-60 deg. Coordinates should also be defined, I assume these are geographic (not geomagnetic) coordinates. I also recommend to show that this is different from other cases, where the strong diffusion is absent. One summary sentence in the main text and a well documented set of cases in the Supplementary material would be good to prove the point and this would also certainly fit in the format of Nature Communications.

The transition occurs near $L=4$, near the typical plasmapause location. As POES crosses the footprint of the plasmapause, it begins to record electrons that may have encountered high chorus wave power, resulting in rapid pitch angle diffusion. We have added a sentence to this effect to the revised manuscript (line 79).

We have now specified the use of geographic coordinates (lines 65 and 100).

This event was shown as it was the best example found in a non-exhaustive search of storms where POES showed flat pitch angle distributions and VAP recorded high chorus wave power, sufficient to cross the strong diffusion limit. We have looked at other events, but did not find others that met our criteria (flat pitch angle distribution, diffusion coefficients $>$ strong diffusion limit, satellites passing similar MLT and L within a small time window when the other two criteria were met). Most events showed evidence of flat pitch angle distributions, but not many showed high wave power. As our intention is only to demonstrate that these conditions can occur, we do not feel that the difficulty of including additional properly analysed case studies in the supplementary material would enhance the manuscript enough to justify their inclusion. A sentence to this effect has been added on line 153.

line 93... The frequency band over which the measured power spectral density was integrated should be mentioned.

We have now mentioned the frequency band over which the measured power spectral density was integrated, from f_{LHR} to $0.5f_{ce}$ (line 109).

line 93 bis...

The B^2 weighted average of wave vector directions can also be easily obtained from the EMFISIS Waves survey mode data, to verify the assumption of a Gaussian wave normal distribution mentioned on line 417. Again, this can be summarized by one sentence in the main text, with a reference to the technique, and some Supplementary material, all of which still fits in the format of Nature Communications.

We have now specified the use of and width of a Gaussian wave normal angle distribution (line 112), as well as included a plot of VAP EMFISIS data from the day of the storm fitted with a Gaussian in the supplementary material fig 3.

After reviewing the wave normal angle data for this period and seeing that it fitted a wider than typical Gaussian, we recalculated the diffusion coefficients in figure 1 with a wider wave normal angle distribution (30deg to 50deg). This only had a noticeable effect on the values of D_{aa} at higher pitch angles. These values are now better justified by the data, but we do not believe that this has change our conclusions. Figure 1B has been updated to reflect the new calculations.

line 93 tris....

Different techniques for estimation of the plasma density exist in the literature and it should be said here which of them is used. A reference would cure it.

We have now specified that we used the plasma density derived from the frequency of f_{UHR} , according to Kurth et al. 2015 (line 109)

line 120 ... Figure 1B shows that the equatorial squared amplitude would need to be at least 30-50 time larger to in order to reach the theoretical strong diffusion limit in a broader interval of pitch angles for 300 keV electrons. I recommend the authors to specifically state what squared amplitudes are they actually using for their calculation. From Fig 1C I can guess that it is probably around 10^5 nT^2 . This leads to chorus amplitudes above 1nT for reaching strong diffusion at 300 keV. Such amplitudes have been observed by Van Allen Probes data but this should also be supported by reference(s). An alternative explanation can be that the interaction takes place at higher latitudes beyond the Van Allen Probes orbit, where the background magnetic field is stronger, modifying the resonance condition.

We have now stated the squared wave amplitude used for the calculation (0.2 nT^2) (line 130). We have also added a brief discussion of higher latitude interactions, as well as a citation for the where this power lies within the observed distribution of lower band chorus wave power (line 144).

line 126 ... I suggest to mention the scaling factor here

We have now specified the scaling factor used (line 161).

line 130 ... One can guess from the context, but a few words to define E_{SD} at the first occurrence of this symbol would make the text better readable.

E_{SD} is now properly defined in the revised manuscript (line 167).

line 158 ... I recommend to explicitly state that scaling factor for 300 keV is higher than for 30 keV (which should also be given here)

We have now specified that a smaller scaling factor has been used, as well as its value (line 181).

line 191... 1D -> 3D ?

The wrong figure panel was referenced, we have now corrected it to say figure 3D.

line 210... I see a discrepancy here: A factor of 1500 is observed between the model diffusion coefficients and the one calculated from measured waves amplitudes. Yet, according to Fig 1B, at 300 keV we still are by a factor of 30-50 below the strong diffusion limit. It means that a factor of at least 45000 against the model is needed to reach the limit. The problem is that in Fig 3D-F the circled symbols for 300 keV start at a factor of 2000. Therefore I have a problem of more than one order of magnitude here. I guess that I'm just missing something in this estimate but I recommend the authors to add an explanation to the text.

The discrepancy arises because the originally stated factor of 1500 between the model coefficients and the calculated coefficients near the loss cone was incorrect. This was actually the factor at 85deg, where figure 3 was plotted, and was quoted mistakenly.

We have now corrected this to specify that there is a factor of 66 between the model coefficients and the calculated coefficients near the loss cone, and a factor of 1500 for near-equatorially mirroring electrons (line 257).

line 417... I recommend to repeat what is the width of the Gaussian distribution in the model

We have now stated the width of the gaussian distribution of wave normal angles here (line 479).

line 430... The initial exponential decay of the phase space density might be better described, it seems inconsistent with PSD=0 at 10 MeV and also with the initial PSDs in fig3.

We have now clarified that the nature of the code results in PSD=0 being treated as PSD=epsilon as log flux is used (line 512).

This also resolves the apparent inconsistency between the initial PSDs and the high energy boundary, as sufficiently small PSD values in the initial conditions are fixed to a floor of PSD=epsilon, matching the high energy boundary condition.

REVIEWERS' COMMENTS

Reviewer #1 (Remarks to the Author):

I read the revision version of the draft and found that authors have addressed my comments well. I suggest the paper to be published.

Reviewer #3 (Remarks to the Author):

Second review report on manuscript

"Electron Acceleration During Strong Pitch Angle Diffusion"

by T. A. Daggitt et al. submitted to Nature Communications

Re-Reviewed by O. Santolik, 4 September 2023

I went through the authors' responses and through the revised version of the manuscript with its supplementary material, and I was pleased to find that the authors took into account nearly all my comments from the first review of this manuscript.

The only small exception is my comment on line 78 of the original manuscript, concerning the transition of the flux ratio from ~ 0 to ~ 1 at latitudes above of 50-60 deg. My comment probably was not clear enough as it contained several separate

points. The authors responded (a) by clarifying that the transition occurs near the typical plasma pause location, where electrons may have encountered high chorus wave power...lines 79-82 <- here and below the line numbers refer to the revised manuscript with highlighted changes; (b) by specifying the use of geographic coordinates ...lines 65 and 100; (c) by explaining that a full statistical treatment of how often chorus wave driven strong diffusion occurs was not performed, and was beyond the scope of this study...lines 153-155.

As concerns (c) I fully agree that a case study is perfectly OK, and a "full statistical treatment" would not fit in this paper. However, this is not what I suggested. I still think that a causal link between intense low latitude chorus and the flux ratio ~ 1 at $L > 4$ (which the paper seems to imply) might be better substantiated by showing an example case, as a supplementary material, where chorus is weak or absent and the flux ratio stays low at $L > 4$.

My understanding of the author's response is that this is impossible, because the flux ratio is ~ 1 at $L > 4$ in most events that they looked at, even in the absence of intense low-latitude chorus.

This would mean that other explanations should exist for cases when the intense low-latitude chorus

is absent. The problem I see here is that these other mechanisms might be also active when intense chorus is present at low latitudes, and, in fact, these other mechanisms may still be the cause of the flux ratio ~ 1 at $L > 4$, instead of chorus.

I recommend the authors to consider clearly stating that the flux ratio is ~ 1 at $L > 4$ in most analyzed cases, even in the absence of intense low-latitude chorus, and adding at least some discussion on this problem.

A small issue: grammar of the text

"which will not have been observed by VAP A in this event" on line 147 looks strange to me, but not being a native English speaker I leave it up to the authors to possibly fix it, if needed.

As both these remaining comments are minor and easy to be considered by the authors without another review

I recommend the paper for publication in Nature Communications.

These minor comments may be taken into account by the authors while preparing the final version of the paper.

+++

Corrected version re-reviewed by O. Santolik, 21 September 2023

While the text in the corrected version of the main manuscript seems to refer to Figure 1, which highlights one hour of Van Allen Probe A data on 17 March 2013 from 21:00 till 22:00 UTC, the

caption of the new Supplementary figure 3 refers to 24 hours of data on 17 March 2013. This includes three orbits of the spacecraft preceding the event, for which the wave vector angles are irrelevant to the presented analysis. I suggest the authors to narrow this time interval to relevant data acquired from 21:00 till 22:00 UTC. The attached quick-look of Level 4 data from Van Allen Probe A shows that the angle θ_k between the wave vector and local field line is still low and mostly below ~ 30 deg in this interval.

Another important aspect of this correction is that a very different wave normal distribution (see the azimuth angle ϕ_k , which was previously used instead of θ_k) doesn't alter the main results of this study. This shows that these results are very robust, nearly independent of the measured wave vector directions. I find this very interesting and I suggest the authors to comment on this robustness briefly (one sentence might be sufficient), for example in the Discussion section of the paper.

As this correction does not alter the main results of this study, conclusion of my review is still the same.

Author's replies to comments

Author's replies in green between each comment where necessary.

REVIEWERS' COMMENTS

Reviewer #1 (Remarks to the Author):

I read the revision version of the draft and found that authors have addressed my comments well. I suggest the paper to be published.

Reviewer #3 (Remarks to the Author):

Second review report on manuscript
"Electron Acceleration During Strong Pitch Angle Diffusion"
by T. A. Daggitt et al. submitted to Nature Communications

Re-Reviewed by O. Santolik, 4 September 2023

I went through the authors' responses and through the revised version of the manuscript with its supplementary material, and I was pleased to find that the authors took into account nearly all my comments from the first review of this manuscript.

The only small exception is my comment on line 78 of the original manuscript, concerning the transition of the flux ratio from ~ 0 to ~ 1 at latitudes above of 50-60 deg. My comment probably was not clear enough as it contained several separate points. The authors responded (a) by clarifying that the transition occurs near the typical plasmopause location, where electrons may have encountered high chorus wave power...lines 79-82 <- here and below the line numbers refer to the revised manuscript with highlighted changes; (b) by specifying the use of geographic coordinates ...lines 65 and 100; (c) by explaining that a full statistical treatment of how often chorus wave driven strong diffusion occurs was not performed, and was beyond the scope of this study...lines 153-155.

As concerns (c) I fully agree that a case study is perfectly OK, and a "full statistical treatment" would not fit in this paper. However, this is not what I suggested. I still think that a causal link between intense low latitude chorus and the flux ratio ~ 1 at $L > 4$ (which the paper seems to imply) might be better substantiated by showing an example case, as a supplementary material, where chorus is weak or

absent and the flux ratio stays low at $L > 4$.

My understanding of the author's response is that this is impossible, because the flux ratio is ~ 1 at $L > 4$ in most events that they looked at, even in the absence of intense low-latitude chorus.

This would mean that other explanations should exist for cases when the intense low-latitude chorus is absent. The problem I see here is that these other mechanisms might be also active when intense chorus is present at low latitudes, and, in fact, these other mechanisms may still be the cause of the flux ratio ~ 1 at $L > 4$, instead of chorus.

I recommend the authors to consider clearly stating that the flux ratio is ~ 1 at $L > 4$ in most analyzed cases, even in the absence of intense low-latitude chorus, and adding at least some discussion on this problem.

This comment has made clear that our previous reply was unclear. When we stated that we observed flux ratios ~ 1 in all the events we looked at, we did not make it clear that we had only looked at storm events. Outside of storms the flux ratio is usually $\ll 1$ at all L shells when there are no chorus waves at low latitudes. Now that we fully understand what the reviewer is suggesting, we have included supplementary figure 2 in the supplementary information, showing a case when chorus wave power is not observed by the VAP-A satellite and low flux ratios are continuously observed by POES. This reinforces the link between chorus waves and flux ratios ~ 1 . We have also mentioned this case in the manuscript at line 114. Some discussion of other mechanisms is included at line 118.

A small issue: grammar of the text
"which will not have been observed by VAP A in this event"
on line 147 looks strange to me, but not being a native English speaker I leave it up to the authors to possibly fix it, if needed.

The sentence has been re-worded to read more naturally, now on line 208.

As both these remaining comments are minor and easy to be considered by the authors without another review I recommend the paper for publication in Nature Communications. These minor comments may be taken into account by the authors while preparing the final version of the paper.

+++

Corrected version re-reviewed by O. Santolik, 21 September 2023

While the text in the corrected version of the main manuscript seems to refer to Figure 1, which highlights one hour of Van Allen Probe A data on 17 March 2013 from 21:00 till 22:00 UTC, the caption of the new Supplementary figure 3 refers to 24 hours of data on 17 March 2013. This

includes three orbits of the spacecraft preceding the event, for which the wave vector angles are irrelevant to the presented analysis. I suggest the authors to narrow this time interval to relevant data acquired from 21:00 till 22:00 UTC. The attached quick-look of Level 4 data from Van Allen Probe A shows that the angle θ_k between the wave vector and local field line is still low and mostly below ~ 30 deg in this interval.

We have narrowed the time interval to the relevant period, and updated the corresponding data and figure in the supplementary information.

Another important aspect of this correction is that a very different wave normal distribution (see the azimuth angle ϕ_k , which was previously used instead of θ_k) doesn't alter the main results of this study. This shows that these results are very robust, nearly independent of the measured wave vector directions. I find this very interesting and I suggest the authors to comment on this robustness briefly (one sentence might be sufficient), for example in the Discussion section of the paper.

We have included a sentence in the discussion section mentioning this result at line 356.

As this correction does not alter the main results of this study, conclusion of my review is still the same.